

# CAV1 alleviated CaOx stones formation *via* suppressing autophagy-dependent ferroptosis

Yuanyuan Yang[1], Senyuan Hong[1], Yuchao Lu[1], Qing Wang[2,3], Shaogang Wang[1] and Yang Xun[1]

[1] Department of Urology, Tongji Hospital, Tongji Medical College, Huazhong University of Science and Technology, Wuhan, Hubei, China
[2] Department of Urology, Guizhou Provincial People's Hospital, Guizhou University, Guiyang, Guizhou, China
[3] Department of Research Laboratory Center, Guizhou Provincial People's Hospital, Guizhou University, Guiyang, Guizhou, China

Corresponding authors
Shaogang Wang,
sgwangtjm@163.com
Yang Xun, tjxyang1993@163.com

## ABSTRACT

**Background:** Calcium oxalate (CaOx) is the most common type of kidney stone, but the mechanism of CaOx stones formation remains unclear. The injury of renal cells such as ferroptosis and autophagy has been considered a basis for stones formation.
**Methods:** We conducted transmission electron microscope (TEM), reactive oxygen species (ROS), malondialdehyde (MDA), glutathione (GSH), and C11-BODIPY analysis to explore whether CaOx could induce autophagy-dependent ferroptosis *in vivo* and *in vitro*. To explore the possible mechanism, we conducted bioinformatic analysis of patients with or without CaOx stones, Western blot and qPCR were used to identify the different genes we found in bioinformatic analysis.
**Results:** In our study, we found that CaOx could induce autophagy-dependent ferroptosis no matter *in vivo* or *in vitro*, which might finally lead to urolithiasis. Bioinformatic analysis of the GSE73680 dataset indicated that the expression of caveolin-1 (CAV1) was higher in control patients than CaOx stone patients, the STRING database indicated that CAV1 might interact with low density lipoprotein receptro-related protein 6 (LRP6), Gene Set Enrichment Analysis (GSEA) showed that the WNT pathway positively associated with the control group while negatively related to the stone group, and LRP6 was the core gene of the WNT pathway. Western blot found that CAV1, LRP6, and Wnt/β-Catenin were decreased in Human Kidney2 (HK2) cells stimulated with CaOx. Furthermore, the WNT pathway was considered to be involved in autophagy and ferroptosis.
**Conclusions:** We presumed that CAV1 could ameliorate autophagy-dependent ferroptosis through the LRP6/Wnt/β-Catenin axis, and finally alleviate CaOx stone formation.

## INTRODUCTION

Urolithiasis is a common disease in urology, affecting approximately 10% of people worldwide, and increasing yearly (*Singh et al., 2021*). The 5-year recurrence rate of urolithiasis is up to 67% (*D'Costa et al., 2019*). Calcium oxalate (CaOx) stones are the most common type of kidney stones, the latest literature indicates that nearly 80% of kidney stones are composed of CaOx (*Khan, Canales & Dominguez-Gutierrez, 2021*). According to the classical theory, hypercalciuria and hyperoxaluria could induce the formation of Randall's plaque which is the origin of stone formation (*Bouderlique et al., 2019*; *Khan & Canales, 2015*). Randall's plaque is an ectopic calcification, it can gradually grow subcutaneously to the renal interstitium and urinary tract, and eventually break through the epithelium, becoming a nidus exposed to hypercalciuria or hyperoxaluria (*Wiener, Ho & Stoller, 2018*). The injury of the renal tubular epithelial has been confirmed as a basis for Randall's plaque (*Aggarwal et al., 2013*; *Liu et al., 2019*). Previous exploration of cell damage caused by hypercalciuria or hyperoxaluria mostly focused on apoptosis and necrosis (*Chaiyarit & Thongboonkerd, 2020*; *Manjunath, Moeckel & Dahl, 2013*). Recently, more and more studies have paid attention to ferroptosis, a new mode of programmed cell death. We wonder whether high level of calcium or oxalic acid could promote ferroptosis.

Ferroptosis is a form of regulated cell death occurring as a consequence of lipid peroxidation. It is an ancient vulnerability caused by the incorporation of polyunsaturated fatty acids into cellular membranes, the cells have developed complex systems that defend against this vulnerability in different environment and cells' sensitivity to ferroptosis is tightly linked to numerous biological processes (*Stockwell et al., 2017*). Ferroptosis is widely involved in many diseases, the occurrence of ectopic calcification diseases such as coronary atherosclerosis is associated with ferroptosis (*Zhou et al., 2020*). Randall's plaque, as another ectopic calcification of the renal papilla, may also be associated with ferroptosis.

Emerging evidence suggests that autophagy can regulate intracellular iron balance and lipid peroxidation as an upstream mechanism of ferroptosis (*Yang et al., 2019*). Autophagy is a self-regulation process in which proteins and organelles are destroyed under the stimulation of external environmental conditions such as stress, but the cell membrane is not destroyed (*Mizushima & Murphy, 2020*). In recent years, more and more studies have been conducted on autophagy-dependent ferroptosis. Mechanisms of ferroptosis induced by autophagy include iron autophagy induced by NCOA4-mediated ferritin degradation (*Hou et al., 2016*), BECN1-SLC7A11 complex induced inhibition of the system Xc⁻-GPX4 pathway (*Kang et al., 2018*), RAB7A-mediated lipid autophagy (*Bai et al., 2019*), and biological clock protein specific autophagy mediated by ARNTL (*Yang et al., 2019*).

There are already many studies associated with autophagy and urolithiasis, our previous studies demonstrated that CaOx could cause the autophagy of cells, then lead to stone formation (*Wu et al., 2021*). However, it is still controversial whether autophagy accelerates or ameliorates stone formation. *He et al. (2021)* have explored that ferroptosis might cause the stones formation but the concrete mechanism remains unclear.

Our study was conducted to confirm the relationship between autophagy-dependent ferroptosis and urolithiasis. We also explored the possible mechanism CaOx inducing

ferroptosis *via* bioinformatic analysis. Through our study, we aim to identify the relationship between CaOx and ferroptosis, and try to shed new light on the mechanism of urolithiasis.

## MATERIALS AND METHODS

### Data collection and bioinformatic analysis

A microarray dataset from NCBI Gene Expression Omnibus (GEO) was collected for kidney stone disease (https://www.ncbi.nlm.nih.gov/geo/): GSE73680 obtained from three kidney regions: normal renal tissue of CaOx stone patients, plaque renal tissue of CaOx stone patients, and renal tissue of control patients. GEO2R (https://www.ncbi.nlm.nih.gov/geo/geo2r?acc=GSE73680) was used to identify the difference expression of genes. Zero value genes in more than 30% of samples were filtered out and the data were log2 transformed. A volcano plot and heatmap were drawn used GraphPad Prism and Network Analyst (*Xia et al., 2013*). GO and KEGG analysis were applied through Omicshare Tools (https://www.omicshare.com/tools/). Protein-protein interactions was explored *via* the STRING database. Gene Set Enrichment Analysis (GSEA) was conducted by Omicshare Tools and the venn diagram was conducted through (http://bioinformatics.psb.ugent.be/research).

### RNAseq analysis

#### RNA isolation and qPCR analysis

Total RNA was isolated from Human Kidney2 (HK2) cells with or without the stimulation of CaOx using RNeasy Mini kit (Qiagen, Valencia, CA, USA). RNA concentrations were measured using a Nanodrop ND-1000 spectrophotometer (Nanodrop Technologies, Wilmington, DE, USA). 2 μg of total RNA was used for reverse transcription to produce cDNA. qPCR was applied following the instruction of Yeasen Biotechnology (Shanghai, China). Primer sequences are provided in Table S1.

### Animal experiment design

Six male SD rats (8 weeks old, 300 g) were purchased from the experimental Animal Centre of Tongji Hospital, Tongji Medical College, Huazhong University of Science and Technology. Rats were acclimatized to the environment of 12 h light/dark cycle for 1 week in a specific pathogen-free animal house. Then they were randomly divided into two groups of three rats each, simple randomization was used through table of random digit. The two groups included control group and the glyoxylic acid group, rats in the glyoxylic acid group were intraperitoneally injected with glyoxylic acid (10.5 mg/ml, 6.66 ml/kg, Macklin, Shanghai, China) every day for 9 days. All rats were given free access to food and maintained in an environment of 25 °C during the experimental period. 7% chloral hydrate (0.7 ml/100 g) was used for anaesthesia. All rats were euthanized after the above treatment for 2 weeks. Then kidney tissues were collected, fixed with 4% paraformaldehyde or frozen at −80 °C for further use. All procedures were approved by the Animal Care and Use Committee of Tongji Hospital, Tongji Medical College, Huazhong University of Science and Technology.

## Transmission Electron Microscopy (TEM) analysis of renal tissues

Rat renal tissues were fixed in 2.5% glutaraldehyde for 2–4 h and dehydrated with graded alcohol. The tissues were imaged under an electron microscope (HT7800; HITACHI, Hitachi, Ibaraki, Japan).

## Preparation of CaOx crystals

First, a buffer was prepared by mixing 2.8 g Tris-Hcl (Macklin, Shanghai, China), 116.8 mg Nacl (Macklin, Shanghai, China), and 200 ml $ddH_2O$ (Yeasen, Shanghai, China). Then we prepared a 0.5 mM $Na_2C_2O_4$ solution with 3.35 mg $Na_2C_2O_4$ (Macklin, Shanghai, China) and 50 ml buffer (prepared above) and a 5 mM $CaCl_2$ solution with 36.7 mg $CaCl_2$ (Macklin, Shanghai, China) and 50 ml buffer (prepared above). Finally, CaOx crystals were collected by mixing 25 ml 0.5 mM $Na_2C_2O_4$ solution and 25 ml 5 mM $CaCl_2$ solution prepared in the previous procedures, then the mixture was centrifuged at 3,000 r for 5 min, and collected the deposition.

## MDA, ROS, GHS and C11-BODIPY analysis of HK2 cells with different treatment

### Cell culture and treatment

The Human Kidney2 (HK2) cell line was purchased from the Type Culture Collection of the Chinese Academy of Sciences (Shanghai, China) and cultured in RPMI1640 medium (Hyclone, Logan, UT, USA) containing 10% fetal bovine serum (FBS) (Gibco, Grand Island, NY, USA) in the ThermoHERAcell150i/240i (Thermo Fisher Scientific, Waltham, MA, USA) with 5% $CO_2$ and a temperature of 37 °C. Cells were exposed to 2 mM CaOx solution for 24 h after they covered 80% of the well.

### MDA, ROS, GHS and C11-BODIPY analysis

Reactive oxygen species (ROS), malondialdehyde (MDA), and glutathione (GSH) levels were identified under the manufacturer's protocols. The detection of ROS was conducted *via* the Reactive Oxygen Species Assay Kit (S00335; Beyotime, China), the MDA and GSH levels were identified through the MDA and GSH detection kit (Nanjing JianCheng Bioengineering Institute, Nanjing, China). The ROS level of renal tissues were evaluated under a Multifunctional full wavelength microplate reader (M200 PRO; Tecan, Männedorf, Switzerland) with a scale of relative intensity of fluorescence (RFU). The MDA and GSH levels were expressed in μmol/g. The results were an average of three independent measurements. After different treatments, cells were stained with 2 μmol/L C11-BODIPY (GC42959; GLPBIO, Montclair, CA, USA) in accordance with the manufacturer's instructions. After 30 min at 37 °C in the dark, the cells were stained with DAPI nuclear stain (Thermo Fisher Scientific, Waltham, MA, USA) for 10min. Then cells were washed with PBS and imaged by a Leica SP8 confocal laser scanning microscope.

### Western blot analysis

HK2 cells with or without the stimulation of CaOx were lysed in RIPA Lysis Buffer with protease inhibitor phenylmethanesulfonyl fluoride (PMSF) and phosphoproteinase inhibitors (Beyotime Biotechnology, Shanghai, China). Then the proteins were collected

and the production estimated *via* the BCA protein Assay Kit (Boster, Wuhan, China) following the manufactor's instructions. Proteins were separated and isolated using sodium dodecyl sulfate-polyacrylamide gel electrophoresis with 5% and 12% for 120 V, 2 h and then transferred onto polyvinylidene fluoride (PVDF) membranes for 200 ma, 90 min. The buffer we used in the procedure of electrophoresis and transmembrane were purchased from Boster (Wuhan, China), and prepared them with ddH$_2$O or methyl alcohol. The PVDF membranes were blocked with 5% bovine serum albumin for 2 h. The membranes were incubated with primary antibodies against LC3 (14600-1-AP, 1:2000; proteintech, Rosemont, IL, USA), β-actin (66009-1-Ig, proteintech, China, 1:4000), BECN1 (Boster, Wuhan, China, 1:2000), P62 (ABclony, Wuhan, China, 1:2000), caveolin-1 (CAV1) (Santa Cruz, USA, 1:2000), LRP6 (Santa Cruz, USA, 1:2000), and β-Catenin (Santa Cruz, USA, 1:2000) at 4 °C overnight. Then they were incubated with secondary antibodies (Servicebio, Wuhan, China,1:5000) at room temperature for 2 h; the proteins were visualized using enhanced developer (Boster, Wuhan, China). The grey value of these proteins were analyzed with Image-Pro Plus (https://www.mediacy.com/imageproplus).

### Flow cytometry

HK2 cells were grown in a 6-well plate with or without the stimulation of CaOx crystals for 24 h. The cells were stained with 10 μmol/L for 20 min at 37 °C. Then the cells were washed three times with the culture medium and collected in tubes. The flow cytometry analysis was applied *via* CytoFLEX S (Beckman Coulter, Brea, CA, USA), the cytoflex channel we chose was FITC. The relative ROS level was measured by CytExpert software.

## Statistical analysis

Statistical analysis for all data was conducted with SPSS 25.0 (SPSS Inc, Chicago, IL, United States). All results were performed independently three times. T-test was used for experiments comparing two groups. All data were shown as mean ± standard deviations (SD), with a *P*-value of <0.05 considered statistically significant.

## RESULTS

### CaOx activated autophagy and ferroptosis both *in vivo* and *in vitro*

Transmission electron microscope (TEM) analysis showed that the number of autophagic vacuoles was significantly higher in rats injected with glyoxylic acid. What's more, mitochondria of the rat renal became smaller, membrane density increased and the cristae severely disrupted. Mitochondria was generally considered as a primary origin of ROS, and ferroptosis was a ROS-dependent regulated cell death. The damage of mitochondria observed by our study indicated that CaOx could induce the autophagy and ferroptosis in the kidney tissues (Fig. 1A). The level of MDA and ROS were higher and GSH was lower in rats injected with glyoxylic acid (Figs. 1B–1D). C11-BODIPY showed that rats with the injection of glyoxylic acid activated ferroptosis (Fig. 1E). Western Blot results showed the expression of LC3I/II was higher in HK2 cells stimulated with CaOx, the expression of BECN1, a key component involved in autophagy was increased in HK2 cells with CaOx

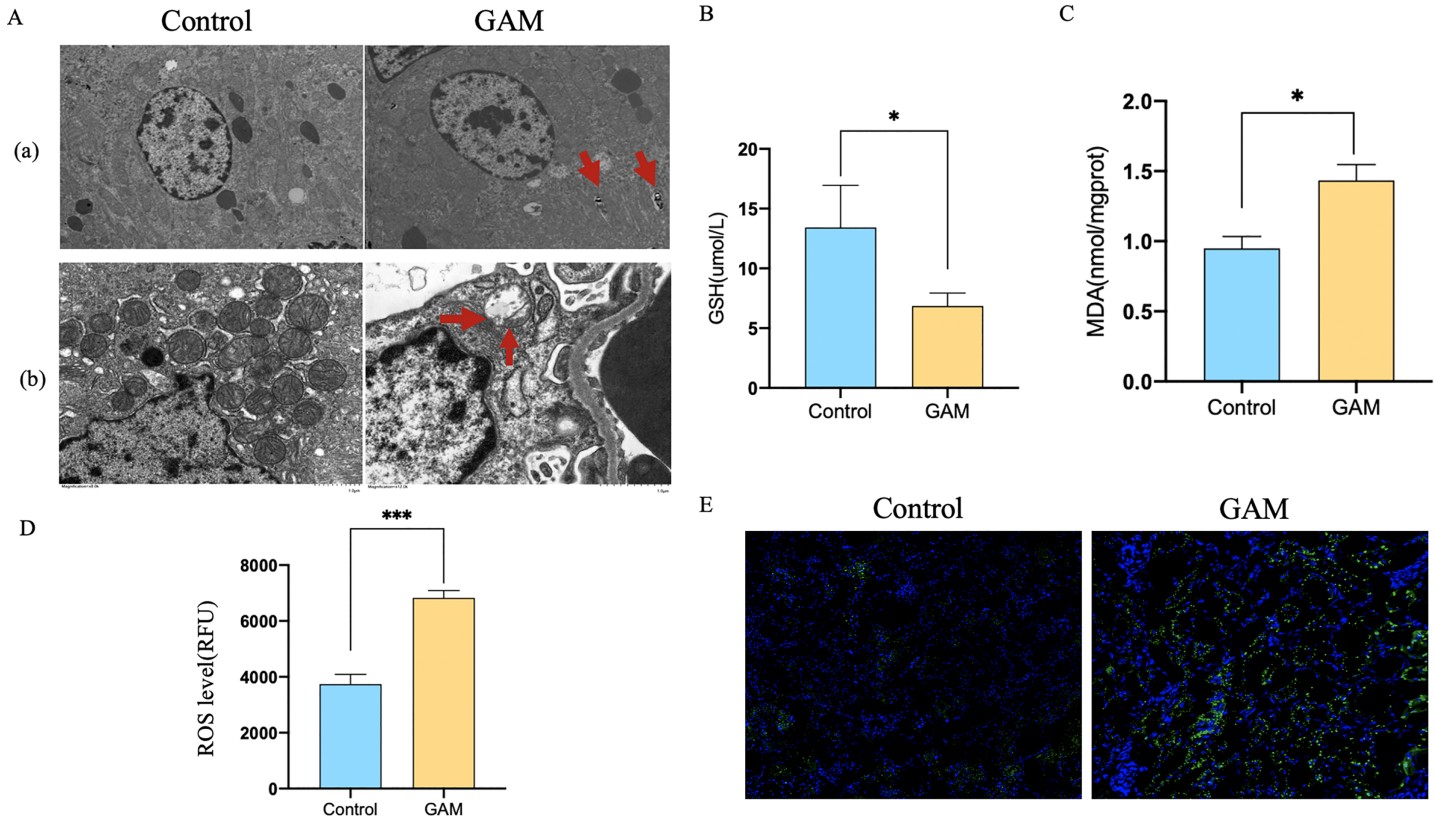

**Figure 1 Rats injected with GAM could induce autophagy-dependent ferroptosis in renal tissues.** (A) Transmission electron microscope (TEM) analysis of rats with or without the injection of glyoxylic acid. (a) Autophagosome was indicated by red arrow. (b) Rats injected with GAM induced ferroptosis in mitochondria. (B–D) GSH, MDA, and ROS level of rats with or without the injection of GAM. The scale of ROS level was RFU. (E) C11-BODIPY showed the level of lipid peroxidation. The data were expressed as mean ± SE, *$P < 0.05$, ***$P < 0.001$ compared with the control group.

stimulation, What's more, the expression of P62/SQSTM1 was decreased in HK2 cells with CaOx stimulation (Fig. 2A). The level of MDA and ROS were higher and GSH was lower in HK2 cells with the stimulation of CaOx (Figs. 2B–2D). What's more, the results of C11-BODIPY indicated that HK2 cells stimulated with CaOx showed more lipid peroxidation (Fig. 2E). These results demonstrated that CaOx could induce autophagy and ferroptosis both *in vivo* and *in vitro*.

## Genes profiling of CaOx and control patients

To investigate the genes associated with CaOx stones formation, we examined the data uploaded to the Gene Expression Omnibus (GEO) database (GSE73680), which performed a microarray analysis for comparing the gene expressions among renal papillary RP and normal tissue of CaOx and normal papillary tissue of control patients. Detailed information of these patients was shown in Table S2. We compared the gene expressions among renal papillary RP and normal tissue of CaOx patients *via* GEO2R,the result showed that there are no significant difference (adj.P.Value > 0.05) between these two groups (Excel S1). We then compared the gene expressions among CaOx and control patients, 243 dysregulated genes were identified significantly ($P < 0.05$), top 50 genes were

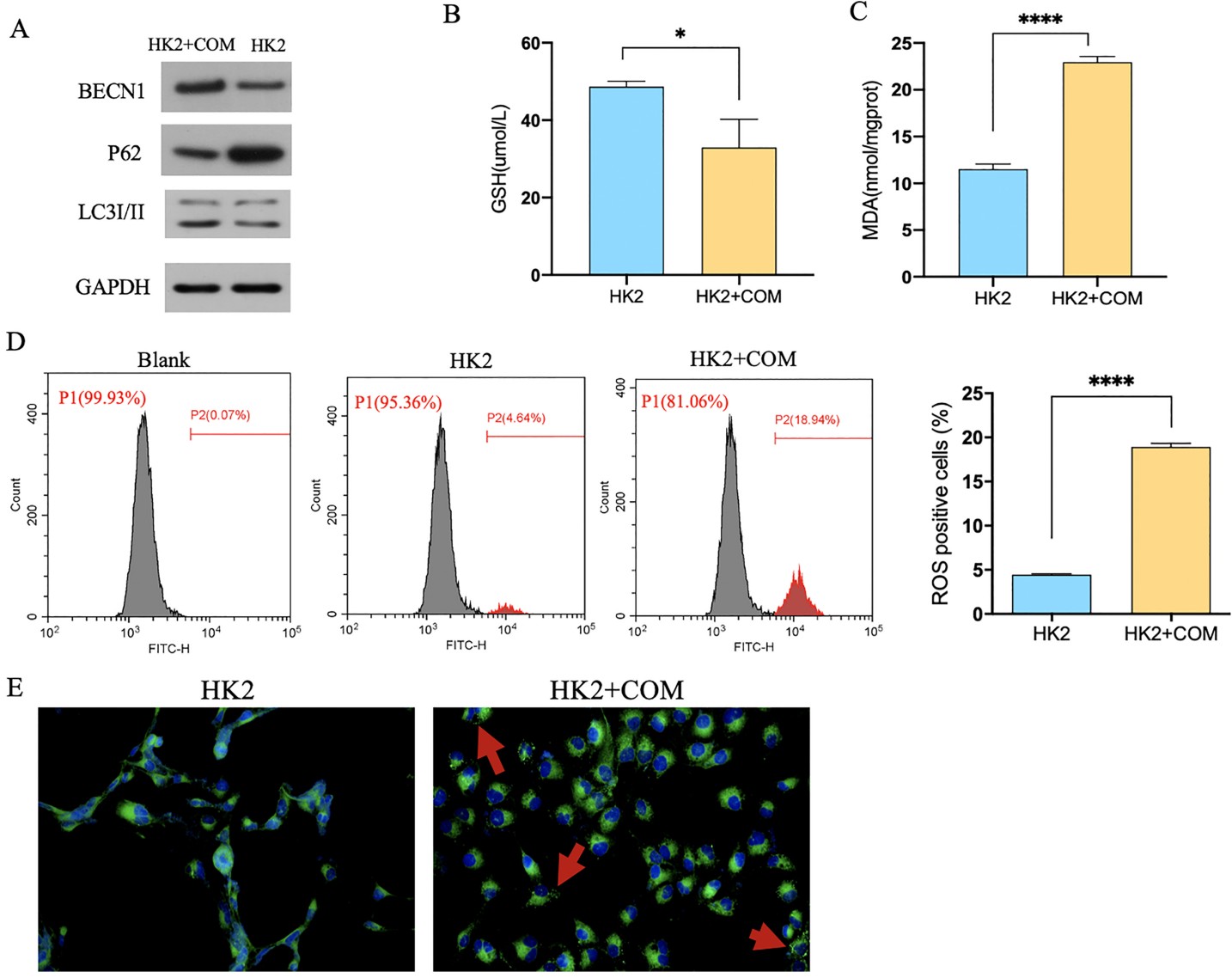

**Figure 2 CaOx could induce autophagy-dependent ferroptosis in HK2 cells.** (A) The results of Western Blot indicated that HK2 cells stimulated by CaOx showed severer autophagy. (B–E) The analysis of GSH, MDA, ROS and C11-BODIPY indicated that HK2 cells with CaOx stimulation undergoing ferroptosis. The data were expressed as mean ± SE, *$P < 0.05$, ****$P < 0.0001$ compared with the control group.

selected to perform in the heatmap (Fig. 3A). When using absolute log2 fold change (log2FC) >2 as the screening criteria, 41 up-regulated genes and five down-regulated genes were identified (Fig. 3B).

## Bioinformatic analysis of dysregulated genes

To explore the possible biological functions of dysregulated genes, we conducted GO analysis using the OmicShare tools. Results demonstrated that these genes were mostly enriched in cellular components (membrane, protein-containing complex), biological processes (metabolic process, cell killing, immune system process), and molecular function
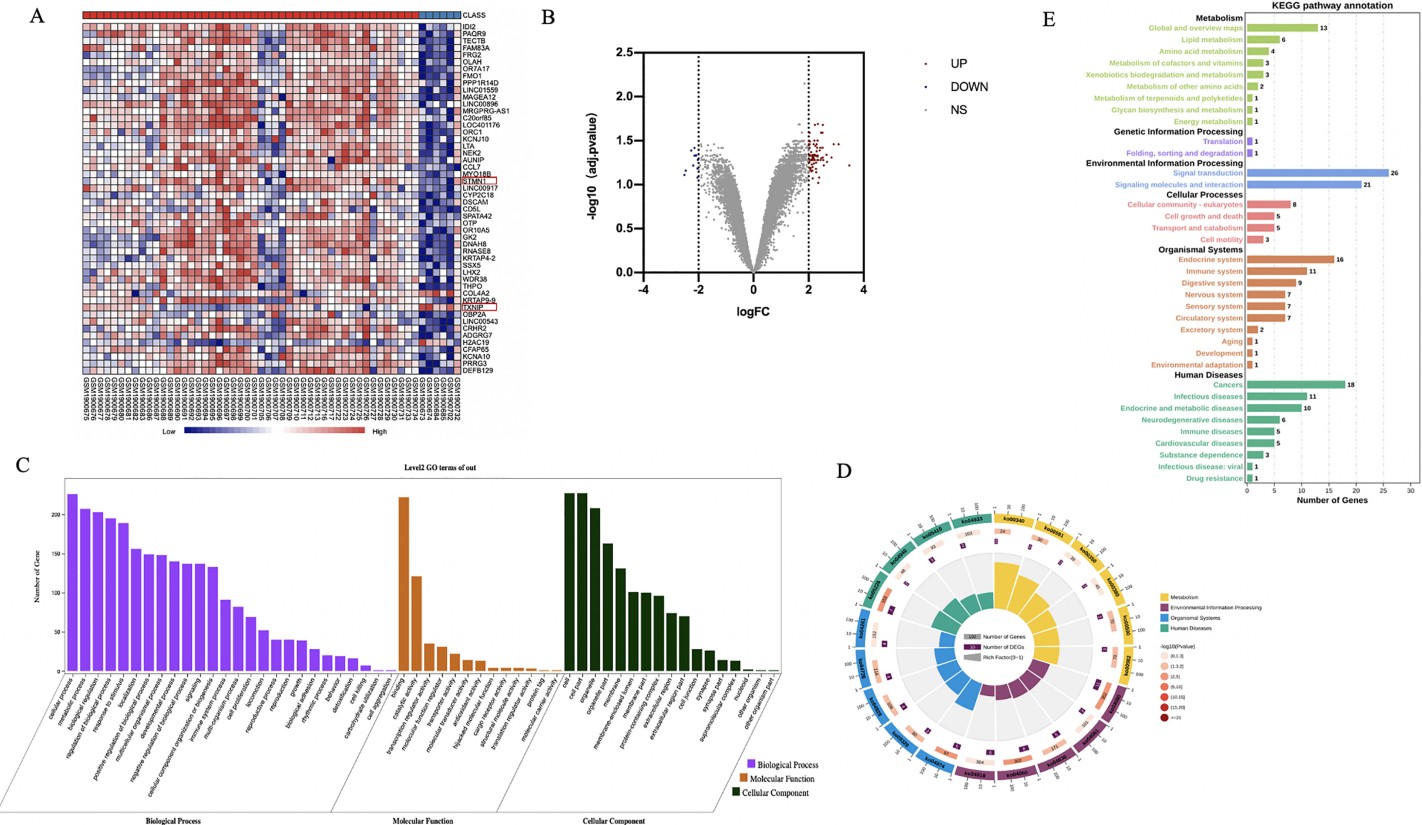

**Figure 3 Profiling of dysregulated genes comparing between CaOx stone and control patients.** (A) With screening criteria of adjusted $P < 0.05$ and $\log_2 FC > 2$ or $\log_2 FC < -2$, 41genes were identified as up-regulated in CaOx groups, while five genes were down-regulated. (B) The top 50 dysregulated genes were shown in this heatmap, the blue bar above represented control groups, while the red represented CaOx groups. (C) GO analysis of dysregulated genes. (D) Circular map of the top 20 enriched pathways of dysregulated genes. (E) KEGG pathway analysis of these dysregulated genes.

(binding, antioxidant activity) (Fig. 3C). While KEGG pathway analysis indicated that dysregulated genes were enriched in lipid metabolism, signal molecules and interaction, cell growth and death, and the immune system (Figs. 3D, 3E). Given that injury and death of cells are the essential basis for the formation of Randall's plaque which is the start of CaOx stones we focused on whether ferroptosis plays some role in stone formation.

## Four dysregulated genes were identified related to ferroptosis

In order to find the relationship between ferroptosis and CaOx stones, we compared the dysregulated genes with genes related to ferroptosis listing in FerrDb (http://www.zhounan.org/ferrdb/legacy/operations/help.html#), we finally found four common genes shared both in Ferroptosis clusters and dysregulated clusters (Fig. 4A). The expression of these four genes were listed in Table 1. Scatter plot showed the expression of these four genes, STMN1, TXNIP and CAV1 were of significant difference between CaOx and control patients (Fig. 4B). To confirm these results *in vitro*, we preformed qPCR of cells with or without the stimulation of CaOx; CAV1 was of high expression in cells without the stimulation of CaOx and it was of significant difference (Fig. 4C). For mechanistic

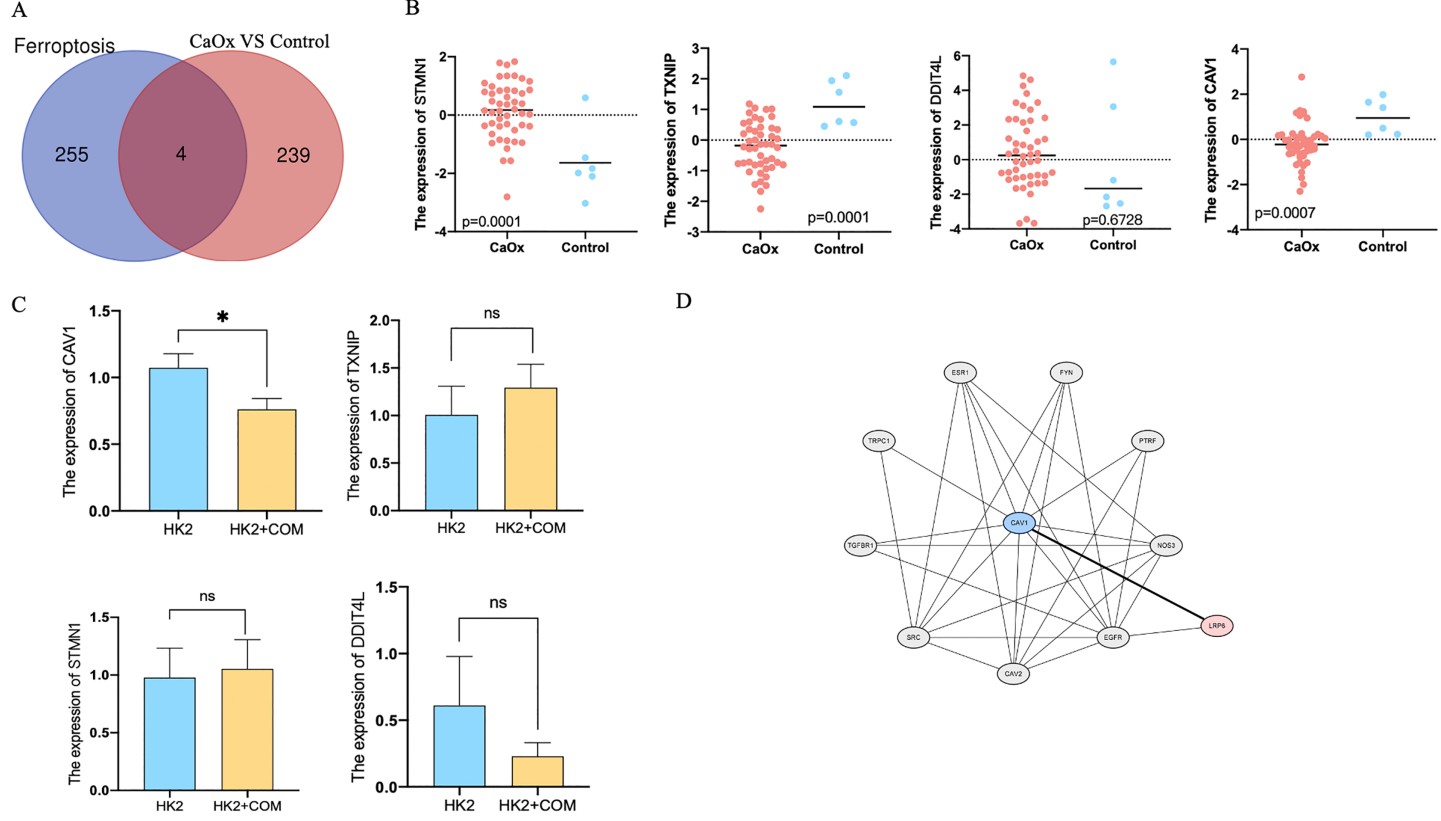

**Figure 4 Four dysregulated genes were identified related to ferroptosis.** (A) A total of 243 dysregulated genes were identified between CaOx stones and control patients, of which four genes were identified involved in ferroptosis. (B) Scatter plot showed the expression of STMN1,TXNIP, DDIT4L and CAV1. (C) The fold change expression of STMN1, TXNIP, DDIT4L and CAV1 in HK2 cells with or without the stimulation of CaOx. (D) The protein-protein interaction network for CAV1. The data were expressed as mean ± SE, *$P < 0.05$ compared with the control group.

**Table 1 Four dysregulated genes.**

| Symbol | LogFC | Adj.P.Val | Name |
|---|---|---|---|
| STMN1 | 1.7999 | 0.032313 | Stathmin 1 |
| TXNIP | −1.4391 | 0.034691 | Thioredoxin interacting protein |
| DDIT4L | 0.444 | 0.72333 | DNA damage inducible transcript 4 like |
| CAV1 | −1.5722 | 0.047168 | Caveolin 1 |

exploration, bioinformatics prediction of the protein network was performed on the STRING database, Low-density lipoprotein receptor-related protein 6 (LRP6) was identified as a downstream protein that interacted with Caveolin-1 (CAV1) (Fig. 4D).

### WNT signaling pathway showed lower expression in CaOx patients

Analyses of the gene signatures of CaOx and control patients *via* gene set enrichment analysis (GSEA) indicated that genes of control patients was correlated with the WNT signaling pathway gene signatures (Fig. 5A). The top 10 core enrichment genes were

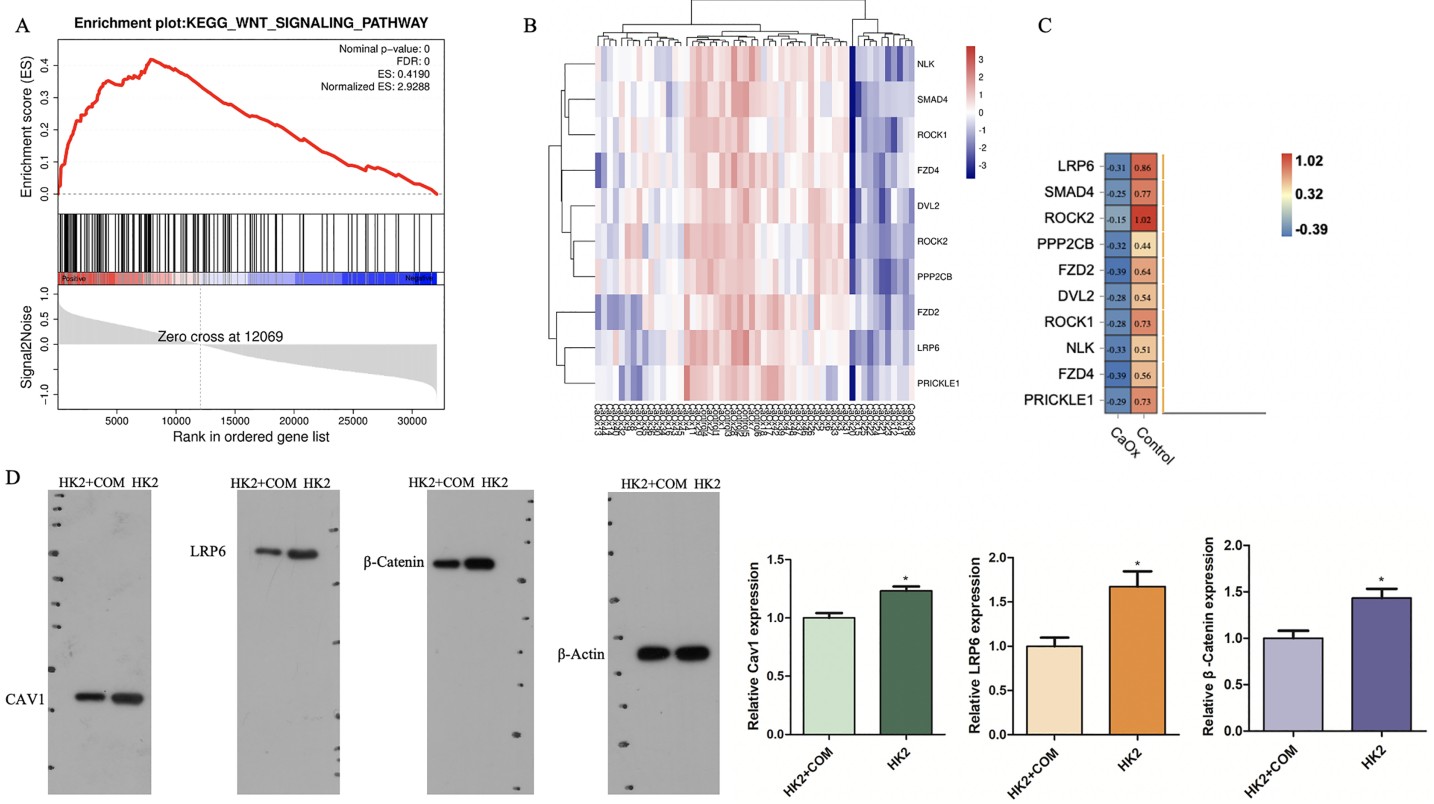

**Figure 5  GSEA analysis of pathways genes derived from CaOx stones and Control patients mostly involved in.** (A) GSEA analyses indicated that Control group was significantly correlated with the WNT signaling pathway in GSE73680 datasets. (B, C) The expression of the top 10 core genes. (D) Western Blot analysis of the expression of CAV1, LRP6 and Wnt/β-Catenin in cells with or without the stimulation of CaOx crystal. The data were expressed as mean ± SE, *P < 0.05 compared with the control group.

shown in a heatmap, of which LRP6 was the most enriched gene (Figs. 5B, 5C). These results suggested that LRP6 might inhibit stone formation *via* the WNT pathway. What's more, some studies have reported that the WNT signaling pathway could suppress ferroptosis *via* high consumption of GSH. We inferred that high expression of the WNT might prevent CaOx stone formation *via* suppressing ferroptosis. To validate that the WNT pathway is involved in the mechanism of ferroptosis, we conducted Western blot analysis to detect the expression of CAV1, LRP6 and Wnt/β-Catenin. The expression of these proteins were increased in HK2 cells without stimulation with CaOx which was consistent with the results of bioinformatic analysis (Fig. 5D).

## DISCUSSION

The prevalence of kidney stones has increased worldwide over the past years, but the mechanism underpinning it remains unclear. Among kidney stones, CaOx is the major mineral constituent of most stones, which makes up to 85% (*Worcester & Coe, 2008*). The major risk factor is an elevation in urine calcium and oxalate. Many studies have shown that elevated calcium and oxalate could induce renal epithelial cell injury (*Wang et al., 2018*; *Qin et al., 2018*). The epithelial cell injury caused by autophagy-dependent ferroptosis has been involved in many diseases, cells have ferroptosis always with ROS
increased, which indicated lipid peroxidation (*Su et al., 2019*). *Khan (2013)* found that ROS level is significantly associated with CaOx stones formation.

In our study, we found that rats injected with glyoxylic acid had autophagy and ferroptosis of their kidneys *via* the observation of TEM. Furthermore, expression of autophagy markers LC3, BECN1, and P62 indicated that the stimulation of CaOx could cause the upregulation of autophagy while the levels of ROS, MDA, GSH and C11-BODIPY in HK2 cells stimulated with CaOx crystals and rats injected with glyoxylic acid also demonstrated the happenings of autophagy and ferroptosis. Previous studies have confirmed that hypercalciuria and hyperoxaluria played important roles in stone formation (*Song & Maalouf, 2020*). However, the mechanism of stone formation remains unclear.

Autophagy has been largely considered an upstream of ferroptosis, which might be induced dependent on the appearance of autophagy (*Yang et al., 2019*). Our previous study showed that hypercalciuria and hyperoxaluria could increase the autophagy of cells and lead to kidney stone formation (*Wu et al., 2021*). In this current study, we found that hyperoxaluria could cause not only autophagy, but also ferroptosis which has not been further explored in urolithiasis. Ferroptosis was found to be important in many diseases, coronary atherosclerosis was considered an ectopic calcification, inhibition of ferroptosis could alleviate coronary atherosclerosis through attenuating lipid peroxidation and endothelial dysfunction (*Meng et al., 2021*; *Yang, Song & Yin, 2021*). So we presumed urolithiasis, a disease based on Randall's plaque might have strong associations with ferroptosis. Therefore, it is of great significance to explore the mechanism of CaOx inducing autophagy-dependent ferroptosis, it might provide new in sights on kidney stones prevention and treatment.

In order to explore the mechanism in depth, we conducted bioinformatic analysis of the datasets by profiling the genes of kidney stone patients and non-stone controls. We found the dysregulated genes were enriched mostly in cellular components (membrane, protein-containing complex) and biological processes (metabolic process, cell killing, immune system process), molecular function binding, antioxidant activity), which indicated metabolism and oxidative stress played important roles in stone formation. This is consistent with previous studies on kidney stone formation; oxidative stress would cause cells damage which was the necessary basis for stone formation (*Jia et al., 2021*). While KEGG pathway analysis indicated that dysregulated genes were enriched in lipid metabolism, signal molecules and interaction, cell growth and death, and the immune system. These demonstrated that ferroptosis played important roles in stone formation for ferroptosis was essentially a kind of lipid metabolism change (*Li & Li, 2020*). What's more, we found that four out of 243 dysregulated genes were enriched in ferroptosis, we then identified the expression of these four genes on HK2 cells with or without the stimulation of CaOx. The expression of TXNIP was found higher in HK2 cells with the stimulation of CaOx when STMN1 was lower in HK2 cells with the stimulation of CaOx. These were different from the expression in the renal tissues we used for bioinformatic analysis and they were not significantly different. However, we found that CAV1 was expressed higher in HK2 cells without CaOx stimulation which was consistent with the results of renal tissues.

CAV1 is known as Caveolin-1, it acts as a scaffolding protein within caveolar membranes and interacts directly with G-protein alpha subunits and can functionally regulate their activity (*Nwosu et al., 2016*). *Lu et al. (2022)* have found that CAV1 could decrease ROS level and inhibit ferroptosis. Furthermore, CAV1 deficiency could promote autophagy *via* improving autophagic flux (*Zhang et al., 2020*). To explore the interaction of CAV1, we searched the STRING database and found that LRP6 and CAV1 could interact with each other. LRP6, also known as Low-density lipoprotein receptor-related protein 6, is a component of the WNT pathway. A previous study indicated that LRP6 could regulate autophagy through the Wnt/β-catenin pathway (*Li et al., 2020*). Furthermore, LRP6 could regulate ferroptosis in cardiomyocytes through autophagy (*Li et al., 2021*). Many studies have demonstrated CAV1 could increase the expression of the Wnt/β-catenin pathway through interacting with LRP6 (*Wang et al., 2020*; *Tahir et al., 2013*).

We also conducted GSEA analysis to find the possible pathway and its core genes that might lead to kidney stones. The results showed that wnt signaling pathway was enriched in control groups compared with stone patients. In our current study, we found that the expression of CAV1, LRP6 and Wnt/β-catenin were decreased in HK2 cells with the stimulation of CaOx which was a validation of our presumption that CAV1 might be a protective factor in autophagy-dependent ferroptosis. The Wnt/β-catenin signal transduction cascade controls myriad pathologies in human, aberrant Wnt signaling underlies many diseases (*Nusse & Clevers, 2017*). *Liu et al. (2021a)* have found that Wnt/GPX4 would suppress ferroptosis with high consumption of GSH. *Liu et al. (2021b)* found that lack of potentiated autophagy flux participated in Wnt/β-catenin pathway regulation, elevated wnt pathway might suppress autophagy. While *Liu et al. (2021a)* have found that the Wnt/β-catenin pathway would suppress ferroptosis *via* upregulation of GPX4 with high consumption of GSH.

We presumed that CAV1 could promote Wnt/β-catenin signaling pathway through up-regulating LRP6, then LRP6/WNT suppressed ferroptosis *via* alleviating autophagy of epithelium and finally decreased kidney stones formation.

This study has certain limitations, it was just a preliminary exploration of how CaOx induced ferroptosis of epithelium, it was an attempt to seek new in sight on prevention and medical treatment of urolithiasis. To our knowledge, we were the first to explore the concrete mechanism of autophagy-dependent ferroptosis caused by CaOx. However, more experimental confirmations still needed to be done. We used only one dataset to conduct bioinformatic analysis which might make the results limited, but we verified it with our own samples at the cellular level to increase the credibility.

Our study identified that CaOx might cause autophagy-dependent ferroptosis *via* down-regulation of CAV1, CAV1 might co-express with LRP6, while LRP6 increased the expression of Wnt/β-catenin pathway, alleviating autophagy-dependent ferroptosis, and finally decreasing stone formation. This theory provided a new in sight on kidney stone formation, and might be used for medical treatment. However, more experimental exploration should be done to confirm our hypothesis.

## CONCLUSION

CAV1 was downregulated in CaOx stone patients and the WNT pathway was positively associated with the control group. We presumed that CaOx could cause autophagy-dependent ferroptosis *via* down-regulation of CAV1, CAV1 might co-express with LRP6, while LRP6 increased the expression of the Wnt/β-catenin pathway, alleviating autophagy-dependent ferroptosis, and finally decreasing CaOx stones formation.

### Funding

This work was supported by the National Natural Science Foundation of China (No. 81974092 and No. 8217032551). The funders had no role in study design, data collection and analysis, decision to publish, or preparation of the manuscript.

### Grant Disclosures

The following grant information was disclosed by the authors:
National Natural Science Foundation of China: 81974092, 8217032551.

### Competing Interests

The authors declare that they have no competing interests.

### Author Contributions

- Yuanyuan Yang conceived and designed the experiments, performed the experiments, analyzed the data, prepared figures and/or tables, authored or reviewed drafts of the article, and approved the final draft.
- Senyuan Hong performed the experiments, prepared figures and/or tables, authored or reviewed drafts of the article, and approved the final draft.
- Yuchao Lu performed the experiments, prepared figures and/or tables, and approved the final draft.
- Qing Wang analyzed the data, authored or reviewed drafts of the article, and approved the final draft.
- Shaogang Wang conceived and designed the experiments, authored or reviewed drafts of the article, and approved the final draft.
- Yang Xun conceived and designed the experiments, authored or reviewed drafts of the article, and approved the final draft.

### Animal Ethics

The following information was supplied relating to ethical approvals (*i.e.*, approving body and any reference numbers):

All research was approved by the Animal Care and Use Committee of Tongji Hospital, Tongji Medical College, Huazhong University of Science and Technology.

## Data Availability

The raw measurements are available in the Supplemental Files.

## Supplemental Information

Supplemental information for this article can be found online at http://dx.doi.org/10.7717/peerj.14033#supplemental-information.

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
