# Peer review of "CAV1 alleviated CaOx stones formation via suppressing autophagy-dependent ferroptosis"

_PeerJ, doi:10.7717/peerj.14033_

## Round 0.1 · original submission · Major Revisions

The manuscript needs a major overhaul prior to be publication ready. Authors might want to submit with necessary changes suggested by reviewers.

Reviewer 1 ·

Basic reporting

• The article is clear and unambiguous, and the Literature references are sufficient to explore the field background/context provided.

• The structure of the article conforms to an acceptable format but the setup of the pages changes, for example, animal experimental design has a layout different from the rest

Experimental design

The aim and scope are well defined Research question well defined

Validity of the findings

All underlying data that have been provided are not robust to sustain the claim that the authors did.
Conclusions are not well stated
For example

1) Need a deep explanation on how the mitochondria membrane density and the number in cistae could be related to the pathology (fig 1)
2) Bodipy is a non-specific marker of the lipid body, is it possible to test this with an IF for the specific marker (fig1E)
3) Only 1 marker of autophagy is not enough to test the hypothesis (fig 2)
4) Missed proof of concept of wnt pathway involved in the mechanism (fig5)

Additional comments

the authors are really clear in defining the aim of the paper, and they have done a good job but they should go deep in more experiments to test their hypothesis.

·

Basic reporting

I have deeply reviewed the manuscript. It is well designed and organized.
Manuscript can be accepted in its present form.

I suggest that you put full abbreviations, for example at page 6 you mentioned about TEM, ROS, MDA, GSH but only short forms, it would be helpful if you put full abbreviations at the beginning.

I suggest changing the resolution of the Figure 3 (A, C.D, E), after zooming in the gene names are not clearly visible. I also suggest highlighting important gene names in the figures, that you described in your manuscript.

I suggest starting the gene names with first letter capital in table1.

I suggest changing the Figure 4D, highlighting the interactions of CAV1 with LRP6 (as you have discussed it in the introduction).

Experimental design

No comments

Validity of the findings

The storyline of the manuscript is novel, emphasizing the unique role of CAV1 in kidney stone formation. Overall, the findings are encouraging.

Reviewer 3 ·

Basic reporting

No clear and professional English used especially in the methodology section

Experimental design

Methods are not described with sufficient detail and information to replicate.
The authors need to follow the high technical and professional standard to draft the manuscript.

Validity of the findings

Some techniques used in the article has not been included in the methodology section.

Annotated reviews are not available for download in order to protect the identity of reviewers who chose to remain anonymous.

---

## Round 0.2 · accepted · Accept

Please proofread the complete manuscript and correct typos and grammatical errors.

Reviewer 3 ·

Basic reporting

No comments except usage of English language. The authors are requested to double check with their language flow throughout the manuscript.

Experimental design

No comments

Validity of the findings

No comments

Additional comments

No comments